# Electromagnetic Scattering Analysis of SHDB Objects Using Surface Integral Equation Method

**Beibei Kong** [1,2] , **Pasi Ylä-Oijala** [1] **and Ari Sihvola** [1,*]

[1] Department of Electronics and Nanoengineering, Aalto University, P.O. Box 11000, 02150 Espoo, Finland; beibei.kong@nmbu.no (B.K.); pasi.yla-oijala@aalto.fi (P.Y.-O.)

[2] Faculty of Science and Technology, Norwegian University of Life Sciences, P.O. Box 5003, 1432 Ås, Norway

[*] Correspondence: ari.sihvola@aalto.fi

**Abstract:** A surface integral equation (SIE) method is applied in order to analyze electromagnetic scattering by bounded arbitrarily shaped three-dimensional objects with the SHDB boundary condition. SHDB is a generalization of SH (Soft-and-Hard) and DB boundary conditions (at the DB boundary, the normal components of the D and B flux densities vanish). The SHDB boundary condition is a general linear boundary condition that contains two scalar equations that involve both the tangential and normal components of the electromagnetic fields. The multiplication of these scalar equations with two orthogonal vectors transforms them into a vector form that can be combined with the tangential field integral equations. The resulting equations are discretized and converted to a matrix equation with standard method of moments (MoM). As an example of use of the method, we investigate scattering by an SHDB circular disk and demonstrate that the SHDB boundary allows for an efficient way to control the polarization of the wave that is reflected from the surface. We also discuss perspectives into different levels of materialization and realization of SHDB boundaries.

**Keywords:** electromagnetic scattering; general linear boundary conditions; numerical analysis; Soft-and-Hard/DB (SHDB) boundary; surface integral equation (SIE)

## 1. Introduction

Boundary conditions are very useful models for defining the field behavior on the boundary of an object. With these conditions, a complex field problem can be significantly simplified, since they avoid considering the fields inside the object. In electromagnetics, the most common examples of boundary conditions are the perfect electric conductor (PEC) and perfect magnetic conductor (PMC) conditions. These conditions state that the tangential component of the electric or magnetic field vanishes on the boundary. The PEC and PMC can be generalized to the impedance boundary condition (IBC), which presents a relation of the tangential electric and magnetic fields with an impedance dyadic [1]. The so-called Soft-and-Hard (SH) boundary [2], and generalized SH (GSH) boundary [3], are special cases of the IBC, as well as the perfect electromagnetic conductor (PEMC) [4].

All of these boundary conditions are expressed in terms of the tangential field components. But it is also possible to formulate boundary conditions for the normal field components. An example is the DB boundary condition which requires the normal components of the electric and magnetic flux densities to vanish on the boundary [5]. Recently, an even more general linear and local boundary condition (GBC) has been introduced which generalizes all above-mentioned boundary conditions [6]. This boundary condition not only has a non-conventional form which contains both the tangential and normal components of the fields, but also exhibits novel electromagnetic properties.

The surface integral equation (SIE) method is a powerful method for numerically analyzing electromagnetic scattering by arbitrarily shaped objects with boundary conditions in an open unbounded

region. The SIE method for the conventional boundary conditions, such as PEC, PMC, and isotropic IBC [7,8], has been well established. The SIE methodology has also been developed for anisotropic IBC [9,10], as well as for DB [11], SH [12], and GSH [13] boundary conditions. In all of these cases, the boundary condition contains either the tangential or normal field components, but not both.

Recently, we have proposed an SIE method for the Soft-and-Hard/DB (SHDB) boundary condition [14]. The SHDB condition is a special case of the GBC. In particular, it can be seen as a generalization of the DB and SH conditions [15]. To our knowledge, this is the first numerical method that has been developed for an electromagnetic boundary condition involving both the tangential and normal field components.

In this paper, we review the SIE method for the SHDB boundary condition and then apply it to a new interesting scattering problem. We demonstrate with numerical experiments that the polarization of the reflected wave from a circular disk can be controlled by adequately modifying the boundary condition on its surface. We conclude the analysis with a discussion on the possible avenues for realizing an SHDB boundary.

## 2. General Linear Boundary Condition

With boundary conditions, electromagnetic scattering and radiation problems can be mathematically formulated as boundary-value problems for Maxwell's equations of time-harmonic fields. In electromagnetics, the so-called general boundary condition (GBC) is the most general linear and local boundary condition [1,6]

$$\alpha_1 \mathbf{n} \cdot c\boldsymbol{B} + \beta_1 \mathbf{n} \cdot c\eta_0 \boldsymbol{D} + \mathbf{a}_{1t} \cdot \boldsymbol{E} + \mathbf{b}_{1t} \cdot \eta_0 \boldsymbol{H} = 0 \tag{1}$$

$$\alpha_2 \mathbf{n} \cdot c\boldsymbol{B} + \beta_2 \mathbf{n} \cdot c\eta_0 \boldsymbol{D} + \mathbf{a}_{2t} \cdot \boldsymbol{E} + \mathbf{b}_{2t} \cdot \eta_0 \boldsymbol{H} = 0. \tag{2}$$

This condition combines the tangential field components with the normal field components on the boundary $S$, the surface of the object. In (1) and (2), $\boldsymbol{E}$, $\boldsymbol{H}$, $\boldsymbol{D}$, and $\boldsymbol{B}$ represent the electric field, magnetic field, electric flux density, and magnetic flux density, $\alpha_1$, $\alpha_2$, $\beta_1$, $\beta_2$ are four dimensionless scalars, $\mathbf{a}_{1t}$, $\mathbf{a}_{2t}$, $\mathbf{b}_{1t}$, $\mathbf{b}_{2t}$ are four dimensionless tangential vectors on $S$, $\mathbf{n}$ is the exterior unit normal vector of $S$, $c = 1/\sqrt{\varepsilon_0 \mu_0}$, $\eta_0 = \sqrt{\mu_0/\varepsilon_0}$, with $\varepsilon_0$ and $\mu_0$, the permittivity and permeability of the background medium.

In this paper, we study the following special case of Equations (1) and (2)

$$T_d \mathbf{n} \cdot c\boldsymbol{B} + T_s \mathbf{a}_t \cdot \boldsymbol{E} = 0 \tag{3}$$

$$T_d \mathbf{n} \cdot c\boldsymbol{D} - T_s \mathbf{a}_t \cdot \boldsymbol{H} = 0. \tag{4}$$

Here, $\mathbf{a}_t$ is a unit tangential vector on $S$, and $T_d$ and $T_s$ are scalar constants. Because the choice $T_d = 0$ and $T_s \neq 0$ returns the SH boundary condition, and $T_d \neq 0$ and $T_s = 0$ leads to the DB boundary condition, Equations (3) and (4) can be regarded as the generalization of the SH and DB boundary conditions. This condition is called the Soft-and-Hard/DB (SHDB) boundary condition [15], and it arises very naturally from the four-dimensional coordinate-free formalism [16].

## 3. Surface Integral Equation Method for SHDB

Let us consider the time-harmonic electromagnetic scattering by a bounded object with the SHDB boundary condition. The time factor is $e^{-i\omega t}$ and the object is assumed to be immersed in a homogeneous lossless medium. The surface equivalence principle [17] is applied to express the scattered electromagnetic fields in terms of the equivalent surface current densities $\boldsymbol{J} = \mathbf{n} \times \boldsymbol{H}$ and $\boldsymbol{M} = -\mathbf{n} \times \boldsymbol{E}$ defined on the surface $S$. For that, we need the Green's function of the background medium, $G$, and the following two surface integral operators [18]

$$\mathcal{T}[\boldsymbol{X}](\boldsymbol{r}) = ik_0 \int_S \boldsymbol{X}(\boldsymbol{r}')G(\boldsymbol{r},\boldsymbol{r}')dS' + \frac{i}{k_0}\nabla \int_S \nabla'_s \cdot \boldsymbol{X}(\boldsymbol{r}')G(\boldsymbol{r},\boldsymbol{r}')dS', \tag{5}$$

$$\mathcal{K}[X](\boldsymbol{r}) = -\int_S \boldsymbol{X}(\mathbf{r}') \times \nabla G(\boldsymbol{r}, \boldsymbol{r}')\, dS'. \tag{6}$$

Here, $k_0$ is the wavenumber of the background medium and $\nabla_s\cdot$ denotes the surface divergence of a tangential vector field, defined on $S$.

Let $\gamma_t \boldsymbol{F}$ denote the tangential component of a vector field $\boldsymbol{F}$ on $S$ and $\mathcal{I}[\boldsymbol{F}] = \boldsymbol{F}$ be the identity operator. Starting from the surface integral representation of the scattered fields, for given incident fields, $\boldsymbol{E}^{\mathrm{i}}$, $\boldsymbol{H}^{\mathrm{i}}$, the tangential field integral equations can be formulated, as follows [19]

$$-\gamma_t \mathcal{T}[\eta_0 \boldsymbol{J}] + \left( \gamma_t \mathcal{K} + \frac{1}{2} \boldsymbol{n} \times \mathcal{I} \right)[\boldsymbol{M}] = \gamma_t \boldsymbol{E}^{\mathrm{i}} \tag{7}$$

$$\left( -\gamma_t \mathcal{K} - \frac{1}{2} \boldsymbol{n} \times \mathcal{I} \right)[\eta_0 \boldsymbol{J}] - \gamma_t \mathcal{T}[\boldsymbol{M}] = \eta_0 \gamma_t \boldsymbol{H}^{\mathrm{i}}. \tag{8}$$

Here, integrals including derivatives of the Green's functions are defined on the surface as principal value integrals, and the wave impedance of the background, $\eta_0$, is used in order to scale the electric current and the magnetic field integral Equation (8). In the following, we use the matrix representation of Equations (7) and (8)

$$\begin{bmatrix} -\gamma_t \mathcal{T} & \gamma_t \mathcal{K} + \dfrac{1}{2} \boldsymbol{n} \times \mathcal{I} \\[2mm] -\gamma_t \mathcal{K} - \dfrac{1}{2} \boldsymbol{n} \times \mathcal{I} & -\gamma_t \mathcal{T} \end{bmatrix} \begin{bmatrix} \eta_0 \boldsymbol{J} \\[2mm] \boldsymbol{M} \end{bmatrix} = \begin{bmatrix} \gamma_t \boldsymbol{E}^{\mathrm{i}} \\[2mm] \eta_0 \gamma_t \boldsymbol{H}^{\mathrm{i}} \end{bmatrix}. \tag{9}$$

As a next step, we add the SHDB boundary conditions to integral Equations (9). In order to allow for combining scalar boundary conditions (3) and (4) with vector integral Equations (9), we modify the original form of the SHDB boundary condition. Because the field integral equations are formulated in terms of $\boldsymbol{J}$ and $\boldsymbol{M}$, we first rewrite Equations (3) and (4), as

$$iT_d \nabla_s \cdot \boldsymbol{M} + T_s k_0 \mathbf{n} \times \mathbf{a}_t \cdot \boldsymbol{M} = 0 \tag{10}$$

$$iT_d \nabla_s \cdot \eta_0 \boldsymbol{J} + T_s k_0 \mathbf{n} \times \mathbf{a}_t \cdot \eta_0 \boldsymbol{J} = 0 \tag{11}$$

using the well-known identities

$$\mathbf{n} \cdot \boldsymbol{B} = \frac{1}{i\omega} \nabla_s \cdot \boldsymbol{M}, \quad \mathbf{n} \cdot \boldsymbol{D} = \frac{1}{i\omega} \nabla_s \cdot \boldsymbol{J}. \tag{12}$$

Subsequently, we transform (10) and (11) to a vector form. Multiplying Equations (10) and (11) with two orthogonal tangential vectors $\mathbf{a}_t$ and $\mathbf{b}_t$ $(= \mathbf{n} \times \mathbf{a}_t)$, and combining the resulting equations together, we obtain

$$\mathbf{b}_t \left[ iT_d \left( \nabla_s \cdot \eta_0 \boldsymbol{J} \right) + T_s k_0 \left( \mathbf{b}_t \cdot \eta_0 \boldsymbol{J} \right) \right] - \left( \mathbf{n} \times \mathbf{b}_t \right) \left[ iT_d \left( \nabla_s \cdot \boldsymbol{M} \right) + T_s k_0 \left( \mathbf{b}_t \cdot \boldsymbol{M} \right) \right] = 0 \tag{13}$$

$$\left( \mathbf{n} \times \mathbf{b}_t \right) \left[ iT_d \left( \nabla_s \cdot \eta_0 \boldsymbol{J} \right) + T_s k_0 \left( \mathbf{b}_t \cdot \eta_0 \boldsymbol{J} \right) \right] + \mathbf{b}_t \left[ iT_d \left( \nabla_s \cdot \boldsymbol{M} \right) + T_s k_0 \left( \mathbf{b}_t \cdot \boldsymbol{M} \right) \right] = 0. \tag{14}$$

Combining the boundary condition Equations (13) and (14) with the integral Equations (9) gives the SIE formulation for scattering by an object with the SHDB boundary condition

$$\begin{bmatrix} -\gamma_t \mathcal{T} & \gamma_t \mathcal{K} + \dfrac{1}{2} \boldsymbol{n} \times \mathcal{I} \\[3mm] -\gamma_t \mathcal{K} - \dfrac{1}{2} \boldsymbol{n} \times \mathcal{I} & -\gamma_t \mathcal{T} \\[3mm] \mathbf{b}_t \left[ iT_d \nabla_s \cdot + T_s k_0 \mathbf{b}_t \cdot \right] & -\left( \mathbf{n} \times \mathbf{b}_t \right) \left[ iT_d \nabla_s \cdot + T_s k_0 \mathbf{b}_t \cdot \right] \\[3mm] \left( \mathbf{n} \times \mathbf{b}_t \right) \left[ iT_d \nabla_s \cdot + T_s k_0 \mathbf{b}_t \cdot \right] & \mathbf{b}_t \left[ iT_d \nabla_s \cdot + T_s k_0 \mathbf{b}_t \cdot \right] \end{bmatrix} \begin{bmatrix} \eta_0 \boldsymbol{J} \\[3mm] \boldsymbol{M} \end{bmatrix} = \begin{bmatrix} \gamma_t \boldsymbol{E}^{\mathrm{i}} \\[3mm] \eta_0 \gamma_t \boldsymbol{H}^{\mathrm{i}} \\[3mm] 0 \\[3mm] 0 \end{bmatrix}. \tag{15}$$

These equations are exactly the same as Equation (22) in [14], on which the theoretical foundations of the present paper are resting. Equation (15) is converted to a matrix equation while using the standard method of moments (MoM) with Galerkin's testing and Rao–Wilton–Glisson (RWG) functions [20]. In this process, the surface of an object is first discretized with planar triangles. Subsequently, both the electric and magnetic surface currents are expanded with the RWG basis functions, and the equations are tested with the same RWG functions. Because the RWG functions are associated with the edges of the mesh, the number of the degrees of freedom of the discretized matrix equation is twice the number of the edges. The matrix due to (15) contains more rows than columns. In other words, it is a non-square matrix and we call (15) the non-square integral equation (NSIE) formulation. The Pseudo inverse of the non-square matrix is calculated to find the unknown coefficients of the basis function approximations of the electric and magnetic currents. Once these coefficients are available, they can be used in order to compute the scattered fields at any point outside the surface.

We note that a square matrix could be obtained by adding the field integral equations and the boundary conditions together. This would simplify the solution of the matrix equation. However, since this formulation has been found to lead to numerically unstable solutions [14], here we only use the NSIE formulation.

## 4. Numerical Results

We consider numerical examples in order to verify the results of the proposed NSIE formulation. We compute electromagnetic scattering by a disk with a surface characterized by the SHDB boundary condition. The diameter of the disk is $3\lambda$ and the thickness is $0.04\lambda$, where $\lambda$ is the wavelength of an incident electromagnetic wave in the background medium. The SHDB boundary condition is defined on the surface of the disk, as follows. Let $(\mathbf{x}, \mathbf{y}, \mathbf{z})$ be three orthogonal unit vectors with $\mathbf{x} \times \mathbf{y} = \mathbf{z}$. On the top and bottom surfaces of the disk, which are parallel with the $x$-$y$ plane, the angle between $\mathbf{a}_t$ and $x$ axis is $\beta$, as shown in Figure 1. On the side surface of the disk, the direction of $\mathbf{a}_t$ is parallel to $z$ axis.

The disk is illuminated by a plane wave with a frequency of 300 MHz. The directions of the incident wave $\mathbf{u}^i$ and the incident electric field $\mathbf{E}^i$ are both parallel with the $x$-$z$ plane, as illustrated in Figure 1. Hence, the $x$-$z$ plane is in the following, called an E plane. The angle between $-\mathbf{u}^i$ and the $z$ axis is denoted by $\theta^i$. The surface of the disk is meshed by planar triangles with an average edge length of $\lambda/10$, giving 6120 edges. The number of the unknowns is twice the number of the edges.

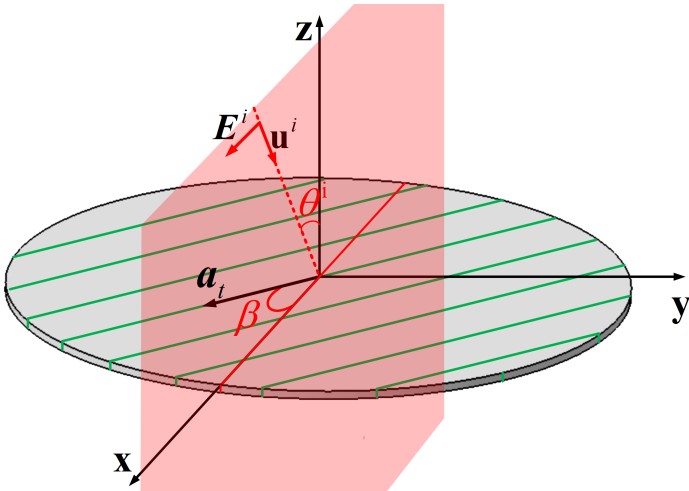

**Figure 1.** A Soft-and-Hard/DB (SHDB) disk with a diameter of $3\lambda$ and a thickness of $0.04\lambda$. The green lines indicate the direction of the vector $\mathbf{a}_t$ on the surface of the disk. Vectors $\mathbf{u}^i$ and $\mathbf{E}^i$ show the directions of the incident wave and the incident electric field.

### 4.1. Special Cases of DB and SH Surfaces

First, we calculate scattering by the disk with the boundary condition parameters $T_d = 1, T_s = 0$ (DB surface) and $T_d = 0, T_s = 1$ (SH surface) and with a normally incident plane wave. The angle $\beta$, which defines the vector $\mathbf{a}_t$, is set to value $20°$. Figure 2 shows the bistatic radar cross sections (RCSs) in the E plane ($x$-$z$ plane). The results for $T_d = 1, T_s = 0$ computed by the NSIE formulation are compared with the results of the SIE method developed for the DB boundary condition [11]. The results of these two independent approaches agree well with each other, as shown in Figure 2.

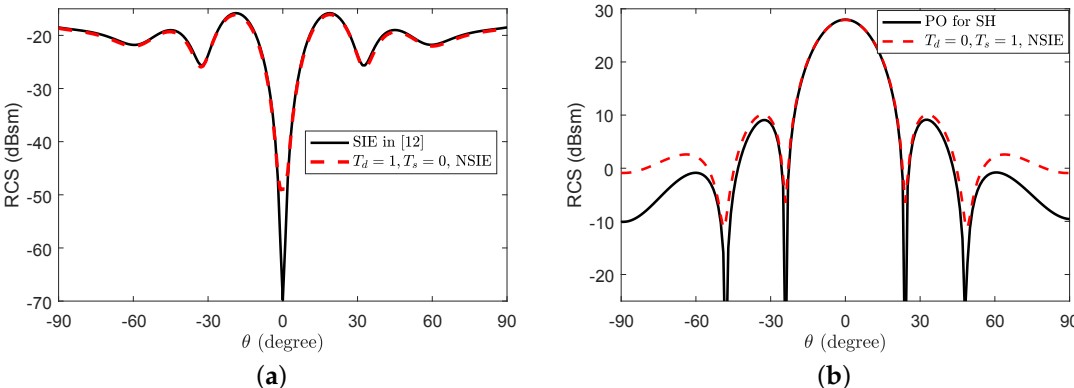

**Figure 2.** Bistatic radar cross sections (RCS) for the SHDB disk shown in Figure 1 at the frequency of 300 MHz with the incident direction $\mathbf{u}^i = -\mathbf{z}$. (**a**) $T_d = 1$, $T_s = 0$ (DB surface). (**b**) $T_d = 0$, $T_s = 1$ (SH surface).

We compare the results of the NSIE with the results of the physical optics (PO) method in order to have an independent solution for verification in the SH case. In the PO method, we employ the reflection dyadics of the SHDB boundary [6] in order to determine the surface currents caused by the wave reflection on the illuminated region. The results of the NSIE and PO agree well, except for $|\theta| > 45°$, as can be seen in Figure 2b. This disagreement on large $\theta$ values is due to the lack of the edge diffraction of the PO method, which makes PO ineffective in providing a reliable benchmark against the numerical NSIE results in this region.

We compare the accuracy of the PO and SIE method in the case of wave interaction with PEC objects in order to provide justification for this reasoning. Figure 3a shows the bistatic RCSs in the E plane of a PEC sphere with a radius of 1.5 m under a frequency of 300 MHz. The results that are obtained by PO and SIE are compared with the analytical Mie solution. It is clear that the accuracy of the SIE method is very good, while the PO method fails to give accurate results. Furthermore, the RCS of a PEC disk in the same case as in Figure 2b is also calculated, and the results are shown in Figure 3b. The results are similar to the deviations in Figure 2b: a discrepancy between PO and SIE for PEC disk is observed when $|\theta| > 45°$. Hence, we have shown that PO fails to provide accurate results in this region, and can conclude that the disagreement between the NSIE and PO results in Figure 2b results from imperfections of the PO method.

Figure 4 displays the electric and magnetic surface current distributions on the top surface of the disk computed with the NSIE. For the DB case, the electric current is parallel with the $x$ axis, and the magnetic current is parallel with the $y$ axis. For the SH case, the electric and magnetic current are both parallel with $\mathbf{a}_t$. We have also verified that by changing the direction of $\mathbf{a}_t$, the distribution and the direction of the surface currents for the DB case will not change, while the direction of the surface currents for the SH case follows the direction of $\mathbf{a}_t$. Thus, we may conclude that the proposed NSIE method gives solutions that agree with our physical intuition of the currents on the DB and SH surfaces.

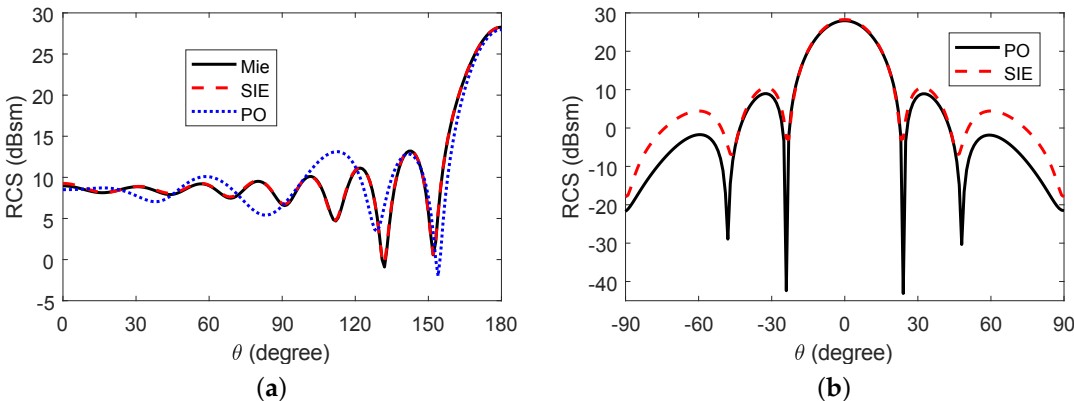

**Figure 3.** Bistatic RCS for perfect electric conductor (PEC) targes at the frequency of 300 MHz. (**a**) Bistatic RCS in the E plane for a PEC sphere with a radius of 1.5λ. (**b**) Bistatic RCS in the E plane for a PEC disk. The size of the PEC disk and the incident field are the same as in Figure 2b.

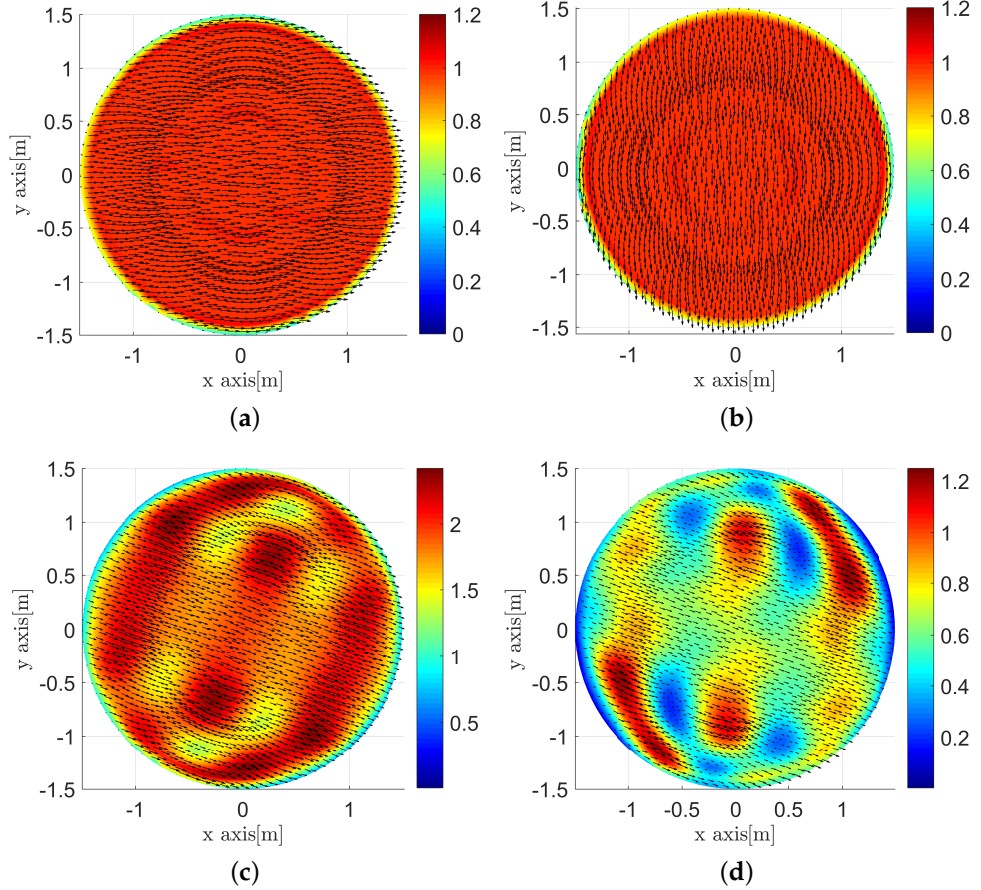

**Figure 4.** Surface current distributions of the SHDB disk shown in Figure 1 at the frequency of 300 MHz for an incident wave with $\mathbf{u}^i = -\mathbf{z}$. (**a**) Real$\{\eta_0 \boldsymbol{J}\}$ for $T_d = 1$, $T_s = 0$ (DB surface). (**b**) Real$\{\boldsymbol{M}\}$ for $T_d = 1$, $T_s = 0$ (DB surface). (**c**) Real $\{\eta_0 \boldsymbol{J}\}$ for $T_d = 0$, $T_s = 1$ (SH surface). (**d**) Real $\{\boldsymbol{M}\}$ for $T_d = 0$, $T_s = 1$ (SH surface).

### 4.2. SHDB Disk, Oblique Incidence Angle

Next we compute scattering by the SHDB disk under oblique incidence. To this end, we first calculate and analyze the co- and cross-polarized reflection coefficients, $R_{co}$ and $R_{cross}$, of an infinite plane. The plane has the same SHDB condition as the top surface of the disk in Figure 1.

Let $(\mathbf{u^r}, \boldsymbol{\theta^r}, \boldsymbol{\varphi^r})$ be a set of orthogonal unit vectors, where $\mathbf{u^r} = \left(\overline{\overline{I}} - 2\mathbf{zz}\right) \cdot \mathbf{u^i}$, $\overline{\overline{I}}$ is the unit dyadic, $\boldsymbol{\varphi^r} = \mathbf{z} \times \mathbf{u^r}/|\mathbf{z} \times \mathbf{u^r}|$ and $\boldsymbol{\theta^r} = \boldsymbol{\varphi^r} \times \mathbf{u^r}$. Splitting the reflected electric field of an infinite planar plane as $\boldsymbol{E^r} = E_v^r \boldsymbol{\theta^r} + E_h^r \boldsymbol{\varphi^r}$, the co- and cross-polarized reflection coefficients are given by [14]

$$R_{co} = \frac{E_v^r}{\|\boldsymbol{E^i}\|} = \frac{(T_s \cos \theta^i \cos \beta)^2 - (T_s \sin \beta + T_d \sin \theta^i)^2}{(T_d + T_s \sin \theta^i \sin \beta)^2 + (T_s^2 - T_d^2)(\cos \theta^i)^2} \tag{16}$$

$$R_{cross} = \frac{E_h^r}{\|\boldsymbol{E^i}\|} = \frac{-2(T_s \cos \theta^i \cos \beta)(T_s \sin \beta + T_d \sin \theta^i)}{(T_d + T_s \sin \theta^i \sin \beta)^2 + (T_s^2 - T_d^2)(\cos \theta^i)^2}. \tag{17}$$

Although the reflection coefficients $R_{co}$ and $R_{cross}$ are for an infinite plane, they can be used in order to evaluate and analyze the numerical results of the PO and NSIE methods for the finite-sized disk that is shown in Figure 1. For the SHDB boundary with $T_d = 1, T_s = 1$, in Figure 5, we plot the values of $R_{co}$ and $R_{cross}$ as functions of $\beta$ under the oblique incidence with $\theta^i = 45°$. From the results shown in Figure 5, we observe that the polarization of a plane wave reflected from the SHDB boundary with $T_d = 1, T_s = 1$ will not change when $\beta = 90°$ and it will be reversed when $\beta = 0°$. We also notice that the co- and cross-polarized reflected waves have the same value as $\beta = 38.7°$.

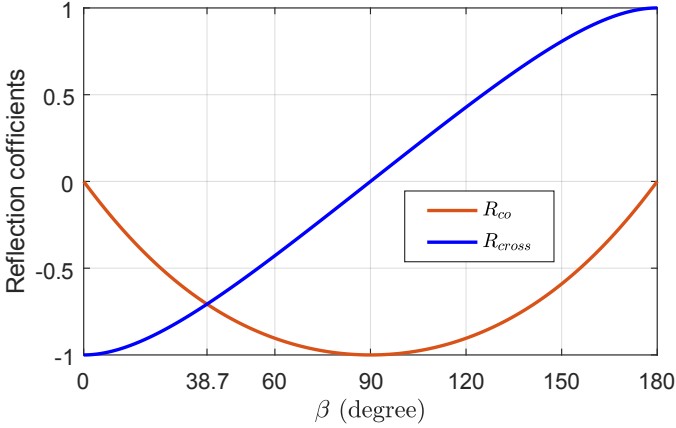

**Figure 5.** Components of the reflection dyadic of an infinite SHDB plane with $T_d = T_s = 1$ for a plane wave incident at $\theta^i = 45°$ as functions of $\beta$.

Next, we calculate the RCS of the disk with $T_d = 1, T_s = 1$, under the oblique incidence with $\theta^i = 45°$. The results shown in the E plane for $\beta = 0°, 38.7°$, and $90°$ are plotted in Figure 6a–c. It can be observed that, as $\beta = 0°$ the cross-polarized RCSs are much larger than the co-polarized ones, while the co-polarized RCSs are much larger than the cross-polarized ones when $\beta = 90°$. In the case $\beta = 38.7°$, the co- and cross-polarized RCSs have the same value at the point of the specular reflection angle $\theta = -\theta^i = -45°$. In Figure 6d, we plot the RCS at $\theta = -45°$, as $\beta$ is increased from $0°$ to $180°$. This result shows a continuous change of the polarization of the scattered field with respect to $\beta$. It is worth noting that, since the reflection dyadic of electric field and magnetic field on the SHDB boundary are exactly the same [14], the polarization of incident field has no effect on the co- and cross-polarized RCS results.

From the results that are shown in Figure 5, we may conclude that the numerical solutions that are computed with the NSIE are qualitatively consistent with the analytical ones for the infinite plane, thus providing additional corroboration for the proposed SIE formulation for objects with the SHDB boundary condition. We also notice that, by rotating the disk, the polarization of the reflected wave can be changed, as desired. This exhibits a potential application of an SHDB boundary, where the disk behaves as a polarization transformer.

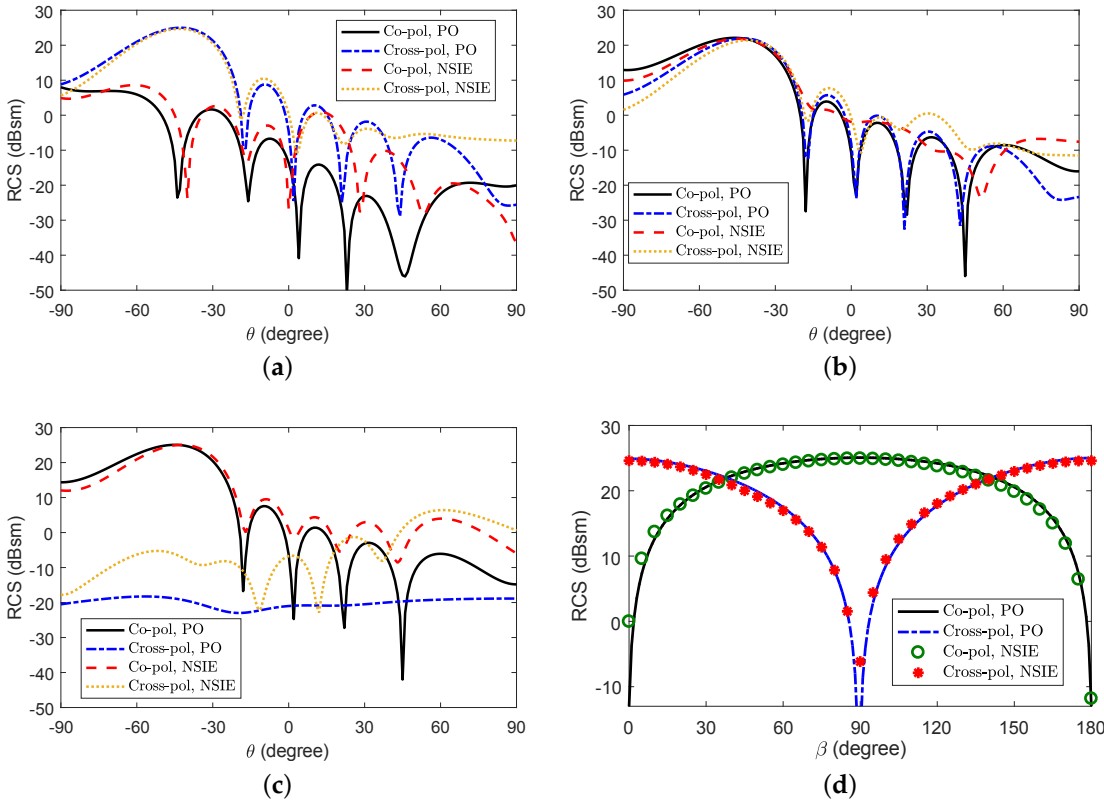

**Figure 6.** Bistatic RCS of the disk in Figure 1 with $T_s = T_d = 1$ illuminated by a plane wave with $\theta^i = 45°$ at the frequency of 300 MHz. The results are shown in the *x-z* plane. (**a**) $\beta = 0°$. (**b**) $\beta = 38.7°$. (**c**) $\beta = 90°$. (**d**) Bistatic RCS at $\theta = -45°$ as a function of $\beta$.

## 5. Perspectives into Realization of the SHDB Boundary

As the computational analysis in the previous section shows, scatterers that are coated with an SHDB boundary offer promising prospects into the modification and transformation of the reflected fields, polarization engineering, and further scenarios of wave manipulation. A natural follow-up question concerns the possibilities to materialize and realize such boundaries and, ultimately, to fabricate surfaces mimicking those.

While the approximate realization of certain boundary conditions in electromagnetics is fairly straightforward (for example, a surface of a good conductor like copper (Cu) can well emulate a PEC over a wide frequency range), a search for the realization of more complex boundary conditions can be much more elaborated. Different levels can be distinguished in the materialization process, as shown in Figure 7.

The surface on which a given boundary condition holds also determines the limits of the spatial domain where the fields are supported. Nothing exists on the other side of this boundary. In contrast to this mathematical idealization, the real-world materialization for the boundary is an interface against a material medium. However, such an interface can only approximately simulate the given boundary condition. This is because a material medium always allows interaction with electric and magnetic fields that penetrate through the interface, although this effect can be attenuated by making the material parameter contrasts extremely large over the interface, like in the case of modeling PEC boundaries by good conductors [21].

The question about materialization of boundary conditions becomes more demanding in the SHDB case, at least for two reasons: as Equations (3) and (4) show, the conditions are (a) anisotropic and (b) they intercouple the tangential and normal components of the fields.

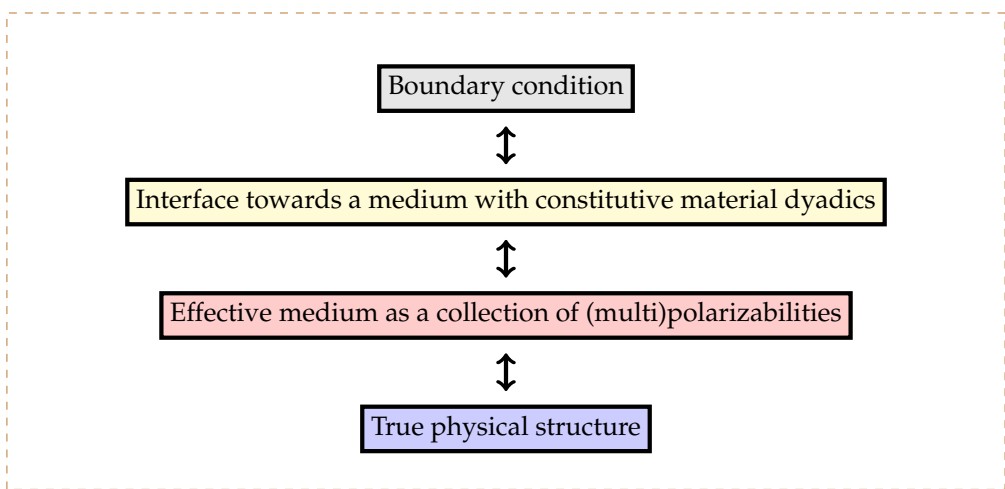

**Figure 7.** Levels of materialization of a given boundary: the medium whose surface imitates the boundary condition is described by a set of electromagnetic constitutive material parameters. On the other hand, the ultimate physical material itself can be modeled as a collection of polarizable particles whose electric and magnetic dipole moments effectively homogenize into these constitutive dyadics.

The SH surface in microwave engineering is a well-known example of realizing an anisotropic boundary [22]. A corrugated conducting surface, where the narrow corrugations are quarter-wavelength deep, acts as a short-circuit for both the electric and magnetic field components in the direction of the corrugations. For example, in optimizing the performance of microwave horn antennas, such a structure has proven to be useful [23,24]. To reach for higher frequencies, another possibility for mimicking an anisotropic boundary is to fabricate a composite in which aligned polarizable inclusions would short-circuit the fields in one of the in-plane directions. The inclusions should be tailored [25] in the manner that they attenuate both the electric and magnetic field components in this direction. This strategy might pave the way towards an SH realization at terahertz and infrared frequencies.

For boundaries forcing restrictions on the normal components of the fields, the fundamental conditions are defined by the DB boundary, in which the normal components of both the electric and magnetic flux densities must vanish. Such conditions are as primary as PEC conditions due to their parameter-free character. The PMC (perfect magnetic conductor) boundary [26] is another example of a condition that does not involve any free parameter. One possible materialization of the DB boundary is a dielectric–magnetic, strongly uniaxially anisotropic medium, where both the axial permittivity and permeability components vanish [27]. A recent experimental metasurface realization of the DB boundary [28] utilizes compact DB-acting elements that are arranged in a regular manner within unit cells, forming a single-layer lattice.

To realize an SHDB boundary—and, in particular, to reach the bottom layer in Figure 7—requires further efforts. It turns out that, unlike in the realization of some of the more traditional boundary conditions, where the effective material parameters can be dielectric, conducting, or magnetic, in the SHDB realization, magnetoelectric coupling is necessary. In other words, the effective medium with which to mimic SHDB condition has to be bianisotropic. The need for bianistropy can be readily seen from the character of the boundary relations (3) and (4) that intercouple electric and magnetic field quantities in a non-conventional manner. The surface of the so-called skewon–axion medium [15] has been shown to reproduce the SHDB boundary conditions. In fact, the skewon–axion medium spans a very wide range of material possibilities: its full description requires 16 scalar parameters [29]. The characterization of an electromagnetic boundary contains less degrees of freedom than what are needed for the constitutive description of a sample of three-dimensional material. Therefore, there can be several different material realizations for a given boundary condition. In [30], a pseudochiral structure has been suggested as another example of bianisotropic materialization of the SHDB surface.

## 6. Conclusions

A surface integral equation (SIE) method is presented for electromagnetic scattering by arbitrarily shaped three-dimensional objects with the Soft-and-Hard/DB (SHDB) boundary condition. In the proposed method, the boundary condition is first expressed in terms of the surface currents, and it is then transformed to a vector form. This form can be combined with the tangential field integral equations and discretized while using the method of moments (MoM) and RWG functions.

The response of a circular disk with the SHDB boundary condition is analyzed in order to verify the performance of the proposed non-square integral equation (NSIE) formulation, and also to study the effect of the SHDB condition on the wave reflection. The results show that the SHDB boundary can be used to manipulate the polarization properties of the wave reflected from it, in a manner that is consistent with the theoretical analysis and results of the physical optics (PO) method.

The particular boundary condition under study (SHDB) is rather complex, but it is not the most general linear and local boundary condition in electromagnetics. On the other hand, the proposed NSIE-based numerical method is expressed in a universal form that can also be applied for more general boundary conditions than SHDB. Another perspective that is related to complex boundary conditions is their realization with existing "ordinary" materials. This aspect was discussed in the framework of different levels of the materialization and the realization of the SHDB boundary.

**Author Contributions:** Conceptualization, B.K., P.Y.-O., A.S.; methodology, B.K., P.Y.-O.; software, B.K.; validation, B.K., P.Y.-O., A.S.; formal analysis, B.K., P.Y.-O., A.S.; investigation, B.K., P.Y.-O., A.S.; writing—original draft preparation, B.K.; writing–review and editing, B.K., P.Y.-O., A.S.; visualization, B.K., A.S.; supervision, A.S. All authors have read and agreed to the published version of the manuscript.

**Funding:** This research received no external funding.

**Conflicts of Interest:** The authors declare no conflict of interest.

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
