# Peer review of "Electromagnetic Scattering Analysis of SHDB Objects Using Surface Integral Equation Method"

_photonics, doi:10.3390/photonics7040134_

Round 1

Reviewer 1 Report

In this article the authors present an in-depth study of a boundary condition that generalizes the DB boundary and the Soft-hard boundary. They present an integral equation approach to characterize the scattering by such objects, and validate their results against other implementations and approximate Physical Optics solutions. The numerical simulations suggest that such electromagnetic boundaries may be useful for polarization transformation. Finally, the authors present a discussion of possible paths to realize the proposed boundary condition. I think the article is complete and will be useful for other researchers interested in related problems. I recommend with no hesitations the article publication in its present form.

Author Response

Our response is in the enclosed file.

Reviewer 2 Report

I find the paper interesting and well-written. The authors show a new numerical implementation of the scattering analysis of materials with SHDB boundary conditions. The results appear accurate. I have some minor comments:

  • It is not clear to me why the authors make use of the conditions in (3) and (4). They adopt two scalar quantities Ts and Td in both equations, which they are able to tune to different values and even give rise to the extreme cases SH or DB. What is the reason why the authors use the same scalar factors Ts and Td in (3) and (4)? Could not they use different scalar values for each term in (3) and (4)? Is it just for simplicity?

  • As regards the resulting implementation, it would be interesting that the authors comment on the numerical stability of the matrix in their numerical tests. My guess is that the provided NSIE formulation may become numerically unstable at low frequencies and for dense-grid meshings (i.e. ill-conditioned matrices). The numerical instabilities associated with the impedance matrices resulting from the RWG discretization of the EFIE or the PMCHWT formulations are very well known. Did the authors check whether the conditioning of the impedance matrix resulting from their NSIE formulation is analogous to the EFIE or PMCHWT RWG-implementation?

Author Response

Our response is in the enclosed file.

Reviewer 3 Report

This is an application paper on the control of polarization of reflected waves from circular disks, by modifying the boundary conditions on the surface of the disk. In my opinion this work fits the MDPI Photonics, however, there are serious flaws that need attention.

1. The basic theory for the SHDB SIE has been published in [14]. In Section 3 the authors call (15) the non-square integral equation (NSIE). What is the difference between (15) of this work and (24) of [14]? In both formulations the matrices have more rows than columns. It should be clearly stated that the present application is based on the fundamental development of [14]. In addition, the part "putting them one on the other as in (15)" in the last paragraph of Section 3 is not clear.

2. In Fig. 2(b) a discrepancy between NSIE and PO is observed. Although a brief statement about PO's deviation is given, still the NSIE is not fully validated for |\theta|>45 deg. An even more serious flaw, however, is that the results between NSIE and PO given in Figs 5(a)-(c) for oblique incidence, disagree. To this reviewer's opinion, the results given in Fig. 5 are questionable. For a workaround to this issue, this reviewer suggests to fully validate the NSIE results in Fig. 2(b) and in Figs 5(a)-(c) with an alternative formulation-based solver for both normal and oblique incidence.

3. Since this is an application paper, the authors should have considered the case where the incident E field is normal to the xz plane (see Fig. 1).

4. In the last paragraph on page 6, it is stated that from the results shown in Fig. 4, it is concluded that the numerical solutions computed with the NSIE are consistent with the analytical ones for the infinite plane. Actually, Fig. 4 only displays the values of (16) and (17); no conclusion can be drawn for the credibility of NSIE.

Some minor points: 'DB' in abstract is not defined. The 'a' in a_t vector in Fig. 1 should appear in bold. The part "... need be not only ..." in the last paragraph of Section 5 needs rephrase.

Based on point No 2, this reviewer cannot recommend publication of this manuscript.

Author Response

Our response is in the enclosed file.

Round 2

Reviewer 3 Report

The authors have carefully addressed my points. The manuscript can be accepted as is.